# Combination of Niraparib, Cisplatin and Twist Knockdown in Cisplatin-Resistant Ovarian Cancer Cells Potentially Enhances Synthetic Lethality through ER-Stress Mediated Mitochondrial Apoptosis Pathway

**DOI:** 10.3390/ijms22083916

**Published:** 2021-04-10

**Authors:** Entaz Bahar, Ji-Ye Kim, Dong-Chul Kim, Hyun-Soo Kim, Hyonok Yoon

**Affiliations:** 1College of Pharmacy, Research Institute of Pharmaceutical Sciences, Gyeongsang National University, Jinju 52828, Korea; entazbahar@gnu.ac.kr; 2Department of Pathology, Ilsan Paik Hospital, Inje University, Goyang 10380, Korea; alucion@gmail.com; 3Department of Pathology, Gyeongsang National University School of Medicine and Gyeongsang National University Hospital, Jinju 52828, Korea; kdcjes@gmail.com; 4Samsung Medical Center, Department of Pathology and Translational Genomics, Sungkyunkwan University School of Medicine, Seoul 06351, Korea

**Keywords:** ovarian cancer, cisplatin, cisplatin resistance, PARPi, niraparib, Twist, lethality

## Abstract

Poly (ADP-ribose) polymerase 1 inhibitors (PARPi) are used to treat recurrent ovarian cancer (OC) patients due to greater survival benefits and minimal side effects, especially in those patients with complete or partial response to platinum-based chemotherapy. However, acquired resistance of platinum-based chemotherapy leads to the limited efficacy of PARPi monotherapy in most patients. Twist is recognized as a possible oncogene and contributes to acquired cisplatin resistance in OC cells. In this study, we show how Twist knockdown cisplatin-resistant (CisR) OC cells blocked DNA damage response (DDR) to sensitize these cells to a concurrent treatment of cisplatin as a platinum-based chemotherapy agent and niraparib as a PARPi on in vitro two-dimensional (2D) and three-dimensional (3D) cell culture. To investigate the lethality of PARPi and cisplatin on Twist knockdown CisR OC cells, two CisR cell lines (OV90 and SKOV3) were established using step-wise dose escalation method. In addition, in vitro 3D spheroidal cell model was generated using modified hanging drop and hydrogel scaffolds techniques on poly-2-hydroxylethly methacrylate (poly-HEMA) coated plates. Twist expression was strongly correlated with the expression of DDR proteins, PARP1 and XRCC1 and overexpression of both proteins was associated with cisplatin resistance in OC cells. Moreover, combination of cisplatin (Cis) and niraparib (Nira) produced lethality on Twist-knockdown CisR OC cells, according to combination index (CI). We found that Cis alone, Nira alone, or a combination of Cis+Nira therapy increased cell death by suppressing DDR proteins in 2D monolayer cell culture. Notably, the combination of Nira and Cis was considerably effective against 3D-cultures of Twist knockdown CisR OC cells in which Endoplasmic reticulum (ER) stress is upregulated, leading to initiation of mitochondrial-mediated cell death. In addition, immunohistochemically, Cis alone, Nira alone or Cis+Nira showed lower ki-67 (cell proliferative marker) expression and higher cleaved caspase-3 (apoptotic marker) immuno-reactivity. Hence, lethality of PARPi with the combination of Cis on Twist knockdown CisR OC cells may provide an effective way to expand the therapeutic potential to overcome platinum-based chemotherapy resistance and PARPi cross resistance in OC.

## 1. Introduction

Ovarian cancer (OC) is the deadliest gynecologic malignancy, responsible for over 50 percent of mortality among all gynecologic malignancies worldwide [1,2]. Currently, cytoreductive operation preceded by combination therapy based on platinum and taxane is used as a typical OC treatment regimen for OC [3]. Despite the high response rate of most patients receiving platinum-based chemotherapy, the majority of patients with advanced OC die from recurrent diseases and resistance to platinum-based chemotherapy. Therefore, an alternative therapeutic strategy to treat patients with recurrent OC is urgently needed.

Recently, several novel strategies for potential personalized therapy have been established, including molecular-targeted medicine (small-molecule inhibitors or antibodies), clinical immunotherapeutic applications and the recognition of synthetic lethal partners [4]. Genetic screening techniques, including computer methods, drug screening, genetic modification with shRNA/siRNA or CRISPR, or a combination of these methods, can recognize synthetic lethal partners [5,6,7]. The alteration of the DNA damage response (DDR) pathway is a predictive biomarker of platinum-based sensitivity in various cancers, including OC [8]. The synthetic lethality interfaces between altered genes and molecules involved in DDR that could be therapeutically exploited to preferentially eradicate cancer cells [9]. Therefore, synthetic lethality could be an alternative therapeutic approach to overcome platinum based chemotherapy resistance.

Poly (ADP-ribose) polymerase 1 (PARP1) is a DNA repair protein that regulates the growth and differentiation of cells by repairing single-strand break (SSB) and double-strand breaks (DSB) of DNA [10]. Inhibition of PARP1 is considered to be the most active and exciting new personalized target therapy for the treatment of OC, especially in those patients with relapsed platinum-sensitive OC [2,11]. PARP1 inhibitors (PARPi), including olaparib (LYNPARZA), niraparib (ZEJULA) and rucaparib (RUBRACA), are recommended for the maintenance of care in patients with recurrent ovarian cancer due to improved benefits and fewer adverse reactions in patients with a full or partial platinum-based chemotherapy response [12,13,14,15,16]. However, PARPi agents also become resistant to other chemotherapy agents, as almost half of BRCA mutated OC patients fai to benefit from PARPi [17,18].

Twist is a member of the core transcription factor helix-loop-helix (bHLH), known to be an epithelial–mesenchymal transition (EMT) master regulator associated with tumor recurrence and chemo-resistance [19,20,21,22,23,24,25,26]. Functionally, Twist was identified as a potential oncogene and our previous studies have identified that Twist also contributed to acquire cisplatin resistance in OC cells [27,28,29]. In several studies, Twist has been linked to the development of resistance to platinum-based chemotherapy and metastasis in cancer [20,30,31,32,33,34,35].

We hypothesized that Twist knockdown cisplatin-resistant (CisR) OC cells sup-pressed DDR and exerted synthetic lethal effect by sensitizing these cells to concurrent treatment of cisplatin and niraparib.

## 2. Results

### 2.1. Twist Expression Is Strongly Correlated with Expression of PARP1 Involved in DNA Damage Response and Repair in CisR OC Cells

To investigate the expression pattern of Twist during the development of cisplatin resistance, we have established two CisR OC cell lines (OV90-CisR and SKOV3-CisR) using intermittent incremental dosing method, starting from a low dose of 10, 20, 40, 80 to 100 µM (10 doses for each term) of cisplatin. Beside this, we generated another 4 subline with the level of platinum resistance in between parental (P) and CisR cell, named G1, G2, G3 and G4 for cisplatin 10, 20, 40 and 80 µm respectively (Figure 1A). Our result revealed that Twist expression gradually increased with increasing intermittent cisplatin doses at 10 to 100 µm which may contribute to the development of cisplatin resistance. In both cancer cell lines, Twist amplification coincided with an increased expression of DNA damage response (DDR) genes, including PARP1 and XRCC1 (Figure 1B and Appendix A). Next, we examined all sublines treated with cisplatin (Cis), niraparib (Nira) and olaparib (Ola) with 20 µM doses, each respectively, for 72 h of treatment followed by 72 h of recovery for clonogenic assay (Figure 1C). The cisplatin and PARP inhibitor (PARPi) sensitivity gradually decreased during the development of resistance (Figure 1C and Appendix A). Both cell lines demonstrated decreased sensitivity toward cisplatin and PARPi (Nira and Ola) in their respective resistant cells (Figure 1D); in other words, CisR OC cells were cross-resistant to PARPi.

### 2.2. The Twist Deficient Cisplatin-Resistant OC Cells Were More Susceptible to Cisplatin and PARPi

To investigate whether Twist regulates DNA repair pathway in CisR OC, we transfected cells with Twist siRNA (siTwist) and siRNA negative control (siNC). Twist knockdown effectively downregulated PARP1 and XRCC1 expression in OV90-CisR and SKOV3-CisR OC cells (Figure 2A).

To examine if knockdown of Twist alters the sensitivity of CisR OC cells to Cis and Nira, we treated both drugs at a various concentrations ranging from 0 to 160 µM for 72 h. Twist-deficient cells demonstrated more susceptibility to Cis and Nira compared with Twist-proficient cells at doses starting from 20 µM and up (Figure 2B,C). Twist-knockdown CisR cells showed a decreased IC_50_ in response to Cis and Nira as compared with Twist-proficient CisR OC cells.

### 2.3. Combination Therapy Exhibits Synergetic Effect with Suppression of DNA Damage Response Capacity and Inducing Apoptosis on Twist Deficient Cisplatin-Resistant OC Cells

Combination drug therapies induced synergistic, additive or antagonistic effects depending on the dosage ratio [36,37,38]. We investigated the dose response of both drugs (Cis and Nira) in three combination ratios of 1:1, 2:1 and 1:2 in the OV90-P and SKOV3-P, siNC OV90-CisR and SKOV3-CisR and Twist-deficient OV90-CisR and SKOV3-CisR cells (Figure 3A). The IC_50_ measurements of three combination ratio after 72 h of incubation with original parental, siNC CisR and Twist-deficient CisR cell lines are summarized in Appendix A. In Twist knockdown CisR cells, the IC_50_ values of Cis+Nira at ratios of 1:1 was significantly smaller than that of the 2:1 and 1:2 ratio in the cell lines studied (*p* < 0.05).

Furthermore, using the Compusyn software program, cell cytotoxicity and quantitative values of the drug interaction combination index (CI) were calculated for Twist knockdown CisR OC cells, where CI < 1, CI = 1, and CI > 1 indicates synergism, additive effect, and antagonism, each respectively [39]. The corresponding CI values have been determined to interpret the value of combinations. The combination of Cis and Nira at the ratio of 1:1 had the highest synergistic value compared to other two ratios of 2:1 and 1:2 at fraction of affected cells value of 0.5, 0.75 and 0.90 in the both Twist-deficient CisR OC cells (Figure 3B). For further experiments, the 1:1 mixture in which the IC_50_ (IC_25_ dosage of Cis plus IC_25_ dosage of Nira) dose of respective drug was used as it showed the lowest CI value in both Twist-deficient CisR OC cells (Appendix A).

Moreover, we investigated the effectiveness of Cis and Nira on cell survival and DDR proteins, XRCC1 and PARP1 in original parental, siNC CisR and Twist Knockdown CisR OC. The Cis and Nira effectively reduced cell survival capacity and suppressed PARP1 and XRCC1 expression in both parental OC cells (Figure 4A,B). However, the cell survival capacity and DDR potential of siNC CisR remained constant even after Cis and Nira treatment (Figure 4C,D).

Established that Twist knockdown reduced cell survival capacity by suppressing DDR proteins, we determined whether a combination of Cis and Nira (Cis+Nira therapy) enhanced the DDR blocking capacity and cell death of Twist-knockdown CisR cells compared to single agents; the Twist knockdown CisR were treated with Cis alone, Nira alone and Cis+Nira therapy. As expected, Cis+Nira markedly decreased cell clonogenicity, where the DMSO group had greater clonal proliferation after 72 h of drug incubation followed by 72 h of recovery time (*p* < 0.05, Figure 5A). Indeed, the Cis+Nira effectively blocked DDR capacity by decreased PARP1 and XRCC1 protein expression level, while the DMSO group had significantly greater expression of DDR genes (*p* < 0.05, Figure 5B).

We also investigated the cell death potential of Cis+Nira by measuring relative cell viability and detecting apoptotic proteins, Bax, cleaved caspase-9 and cleaved caspase-3. Additionally, we did anti-apoptotic proteins, Bcl-2. The Cis+Nira significantly reduced relative cell viability compared to the DMSO group (*p* < 0.05, Figure 5C). The Western blot data revealed that Cis+Nira markedly increased cell death by inducing Bax, cleaved caspase-9 and cleaved caspase-3, and reducing Bcl-2 proteins expression level (*p* < 0.05, Figure 5D).

### 2.4. Combination Therapy Is Considerably Effective against Three-Dimensional Cultured of Twist Knockdown CisR OC Cells

The monolayer culture conditions usually do not replicate the in vivo environment [40,41]. Thus, we employed a 3D spheroids culture to recapitulate the organoid traits of the in vivo environment (Figure 6A). The DMSO treated siNC transfected microspheroids progressively increased in size with tight aggregate spheres formation, while DMSO treated siTwist transfected microspheroids increased in size with loosely aggregate sphere formation (Figure 6B). The Cis+Nira therapy remarkably reduced sphere formation demonstrating significantly lower cell viability (*p* < 0.05, Figure 5C). These findings suggest that the Cis+Nira therapy could potentially eliminate recurred CisR OC cells in vivo.

### 2.5. Cancer Cell-Death by Cisplatin and Niraparib Accomplished through Boosting ER Stress-Mediated Mitochondrial-Depended Apoptosis on Three-Dimensional Cultured of Twist Knockdown Cisplatin-Resistant OC Cells

To determine how Endoplasmic reticulum (ER) stress effects on apoptosis following Cis+Nira therapy in 3D cell culture, the protein expression of several ER stress markers was examined by Western blotting. The release of ER chaperons and activation of ER resident proteins enhancing the expression of CHOP are usually thought to be associated with activation of ER stress [42,43,44]. In the present study, the protein levels of GRP78, calnexin, cleaved ATF6, and CHOP were markedly elevated after the treatment of Cis alone, Nira alone or Cis+Nira therapy (Figure 7A,B).

Mitochondrial changes are critical for apoptosis where disturbance in mitochondrial transmembrane potential is one of the earliest intracellular actions to occur following induction of apoptosis [45,46,47,48]. It is identified that mitochondrial outer membrane proteins, Bax (the pro-apoptotic proteins) and Bcl-2 (anti-apoptotic) and inner protein, cytochrome *C* play a vital role in triggering caspase-dependent apoptosis [47,48]. The mitochondrial apoptosis kit employs Mitostain, a cationic dye that produce fluorescence to differentiate healthy and apoptotic cells. Our result demonstrated that red/green ratio reduced by DMSO (siNC), DMSO (siTwist), Cis (siTwist), Nira (siTwist), and Cis+Nira (siTwist) in Twist-knockdown OV90-CisR cells were 8.25 ± 0.94, 6.28 ± 0.76, 4.42 ± 0.47, 3.61 ± 0.40 and 1.63 ± 0.186, respectively (Figure 8A). The values were identical to those observed in SKOV3-CisR (9.85 ± 0.73, 7.01 ± 0.72, 4.63 ± 0.81, 3.99 ± 0.43, 2.24 ± 0.42) (Figure 8B). By comparison, triple combination of Twist-knockdown, Cis, Nira treatment showed the greatest decrease of mitochondrial membrane potential—four to five-times lower than that of the DMSO (siNC) group. Triple combination treatment caused greater mitochondrial damage in both OV90-CisR and SKOV3-CisR cell lines, as indicated by the decrease in mitochondrial membrane potential. Most importantly, triple combination therapy significantly upregulated Bax in 9the mitochondrial fraction (Appendix A) and upregulated cytochrome *C* in the cytosolic fraction (Appendix A) in Twist-knockdown CisR OC cells. In contrast, downregulation of Bcl-2 and cytochrome *C* in the mitochondrial fraction was observed with triple combination therapy (Figure 9 A and Appendix A). The ratio of cytochrome *C* in cytosol to that in mitochondria was significantly higher in the group of combination therapy due to the release of cytochrome *C* from mitochondria to cytosol (Figure 9B). These findings suggest that the efficacy of Cis and Nira on Twist-Knockdown CisR OC cells is associated with decreasing the mitochondrial potential, followed by the translocation of apoptotic protein Bax from cytoplasm to mitochondrial outer membrane with diminishing Bcl-2 activity to facilitate the cytochrome *C* release from mitochondria to cytoplasm.

### 2.6. The Synthetic Lethality of Cisplatin and Niraparib Consists of Expressing Apoptotic Marker and Suppressing Cell Proliferative Marker on Three-Dimensional Cultured of Twist Knockdown Cisplatin-Resistant OC Cells

We applied hematoxylin and eosin (H&E) staining and immunohistochemistry to investigate histological features together with markers of cell proliferation and apoptosis (Figure 9C). Proliferative activity was examined by anti-Ki-67 and apoptosis potential was represented by anti-cleaved caspase-3 immunohistochemistry. The H&E staining of 3D spheroids after generation exhibited tumor cell characteristics of high nucleus to cytoplasm ratios. The siNC transfected spheroids were densely arranged in tight balls. The siTwist transfected spheroids treated with Cis alone, Nira alone or both combinations of Cis+Nira showed developed vascular structures with loose arrangements of discohesive OC cells. Immunohistochemically, Cis alone, Nira alone or combinations of Cis+Nira showed lower ki-67 activity along with higher cleaved caspase-3 immunoreactivity.

## 3. Discussion

In this study, we demonstrated that the Nira and Cis proficiently killed Twist-knockdown CisR OC cell via induction of apoptosis under both in 2D and 3D cell culture conditions. Additionally, through in-detail in vitro experiments, we investigated the potential mechanisms underlying the synthetic lethality of Nira and Cis on Twist-deficient CisR OC cells and provided a schematic representation of the mechanisms and interactions in Figure 10. We found that one of the mechanisms of synthetic lethality is through blocking DNA repair, i.e., suppression of PARP1 and XRCC1 activation. Another important mechanism is the ER stress-mediated mitochondrial apoptosis through boosting release of ER chaperons, GRP78 and calnexin, and activation of ER resident protein, cleaved ATF6 with enhanced expression of CHOP. Mitochondrial outer membrane proteins, Bax and Bcl-2, and inner protein, cytochrome C play a vital role in triggering caspase-dependent apoptosis [47,48,49]. Hence, Twist-knockdown of CisR OC cells greatly improved the translocation of Bax to the outer mitochondrial membrane with reduced activity of Bcl-2, and thus the release of cytochrome C from mitochondria to cytoplasm, leading to irreversible cell death.

Cancer cells are often considered to have abnormalities in DDR including defects in cell cycle checkpoints and/or DNA repair [18,50]. Cancers with DDR impairment demonstrated vulnerability to genome instability by “synthetic lethality” targeting chemotherapy [18,51,52]. Synthetic lethality between DDR pathways has provided a standard for cancer therapy by targeting DDR. PARP1 is an essential cofactor in DDR by incorporating DNA repair effector proteins, such as XRCC1, into the cisplatin-induced DNA injury site and coordinating their action of DNA single-strand break repair [53,54,55,56]. The most notable event of DDR directing mediators is the capability of PARP1 repression as a regulated, synthetic lethal approach, demonstrated in both in vitro and in vivo studies [57,58,59,60,61,62]. Although FDA approved PARPi includes rucaparib, olaparib, talazoparib and niraparib were hypothesized to be effective in a wide range of patients, only small percentage of cancer patients receiving benefit from it due to cross-resistance to other chemotherapeutic drugs [63,64,65,66].

In patients with OC, Twist expression forecasts poor prognostic outcomes, indicating that Twist can play a critical role in OC development [67,68]. Several studies have indicated that Twist expression could be a useful predictor of unfavorable prognosis for OC [69,70]. However, little information is available as to the role of Twist in platinum resistance of OC. In this study, we demonstrated that CisR OC cells exhibit higher expression of Twist with amplification of DDR proteins, PARP1 and XRCC1. We confirmed that both Cis and PARPi sensitivity gradually decreased during the development of resistant cell lines, indicating CisR cells were also cross-resistance to PARPi. Although the knockdown of Twist in CisR OC cells reportedly blocks DDR genes activity, we confirmed that Twist-knockdown sensitizes OC cells to Cis and Nira confirmed by the significantly lowered IC_50_ value.

More importantly, we observed synergetic drug interactions between Cis and Nira in both Twist-knockdown CisR OC monolayer cell culture. Treatment of Nira causes DNA injury and cellular damage by restricting PARP enzymatic action and enhancing the development of PARP-DNA clusters [71,72]. Similarly, Cis induces DNA damage by generating monoadducts and interstrand crosslinks leading to inhibition of DNA and ultimately to cell death [56,73]. The Cis+Nira therapy significantly reduced cell growth rate, and increased the effects of DNA damage and cell death compared with either monotherapy. Based on these results, we believe that knockdown of the Twist with PARPi could be a highly effective strategy in overcoming CisR in OC.

The 3D cell culture aims at recapitulating in vivo tissue architectures to provide physiologically relevant models to study normal development and disease [74,75]. Alt-hough the 3D culture model cannot fully recapitulate all aspects of the in vivo environment, it offers more benefits than a 2D culture. Our study revealed that Cis+Nira therapy with Twist-knockdown considerably decreased the formation of the micr-organoids in 3D cell culture.

ER stress, measured by the magnitude and length of ER stress, is important for both pro-survival and pro-apoptotic functions. It has been a possible target for developing potential drug candidates that regulate specific signaling pathways to either suppress tumor or overcome chemotherapy resistance [76]. Recently several chemotherapy candidates have been investigated in connection to ER stress, which could directly or indirectly affect cancers [77]. Among them, ER stress inducing agents that activated CHOP-GADD34 axis is a promising anti-cancer approach [78,79,80,81]. Our study showed that after Cis and Nira treatment, potential levels of GRP78, calnexin, cleaved ATF6 and CHOP were simultaneously elevated in 3D spheroid cell culture. Although the underlying basis influencing ER stress- mediated apoptosis have not been fully elucidated, growing evidence indicates that mitochondria and ER work together to trigger cell damage signal [82]. The protein levels of pro-apoptotic Bax elevated, whereas anti-apoptotic Bcl-2 dropped after Cis+Nira therapy in Twist-knockdown CisR OC cells. These two proteins cause the external mitochondrial membrane to be permeabilized and consequently release cytochrome C to the cytosol and eventually activating caspases-3, as reflected in our research.

In addition, the IHC examination of Ki-67 revealed downregulated proliferation and increased cleaved caspase-3 expression after combination therapy in Twist knockdown 3D spheroid OC cell culture.

## 4. Materials and Methods

### 4.1. Cell Lines

Two human ovarian cancer (OC) cell lines, OV90 and SKOV3 were obtained from Korean Biotech Co., Ltd., Seoul, Korea, the domestic distributor of American Type Cul-ture Collection (ATCC). The cells were cultured in their respective ATCC recommended growth supplemented with fetal bovine serum (FBS). Two CisR OC cells, OV90/CisR and SKOV3/CisR were generated by the stepwise increment of cisplatin dosing method described in our previous publication [27].

### 4.2. siRNA Transfection

Small interfering RNA (siRNA) transfections were performed according to the manufacturer’s instructions using Lipofectamine 2000 RNAiMAX transfection reagent (Invitrogen). Twist siRNA (#sc38604, Santa Cruz Biotechnology, Dallas, TX, USA) and control siRNA (#sc37007, Santa Cruz Biotechnology, Dallas, TX, USA) were used to generate Twist knockdown (siTwist) and negative control (siNC) cells. The lyophilized siRNA duplex was reconstituted in RNase-free water to create 10 μM stock solutions. The transfected cells were confirmed by Western blot analysis.

### 4.3. Measurement of Cell Viability and IC_50_

Cell viability and IC_50_ were determined using EZ-cytox cell viability kits (#EZ-1000, DLS-1906, DoGenBio Co., Ltd., Seoul, Korea). The cells (1 × 10^4^ cells/well) were plated in 96-well plates and incubated for desired period of time. The cells were treated with different concentration of drugs. After incubation, EZ-cytox solution was added to each well and incubated for 2 h. Absorbance was then recorded on a microplate reader (Synergy H1; BioTek Instruments, Inc., Winooski, VT, USA) at the wavelength of 490 nm. The IC_50_ values were analyzed using GraphPad Prism software (version 5.0, GraphPad Software Inc., San Diego, CA, USA).

### 4.4. Colony Formation Assay

The clonogenic assay was employed for the determination of cell growth using previously described method [83]. In brief, the cells were cultured plates at low density (~500 cells per well) in 12-well for 24 h and then treated with drugs for 72 h followed by 72 h recovery. The plates were then washed with PBS and stained with 0.1% crystal violet solution. The cells were washed until no stain was visible, air dried, and photographed. The dye was extracted using 1% sodium dodecyl sulfate (SDS) solution by continuous shaking and then quantified using a spectrophotometer at 570 nm.

### 4.5. Generation of Three Dimensional (3D) Spheroidal Model

To generate in vitro three–dimensional (3D) spheroidal model utilized combination of hanging drop and hydrogel scaffolds modified methods [27,84]. The details of generation of three dimensional (3D) spheroidal model are described in the supporting information, Appendix A.

### 4.6. Mitochondrial Apoptosis Staining

Mitochondrial transmembrane potential was evaluated by using mitochondrial apoptosis staining kit (#PK-CA577-K250-25, Promo Kine, Heidelberg, Germany), according to manufacturer instruction. The details of mitochondrial apoptosis staining are described in the supporting information, Appendix A.

### 4.7. Mitochondrial and Cytosolic Separation

Mitochondrial and cytosolic fractions were isolated as described earlier [43]. The details of mitochondrial and cytosolic separation are described in the supporting information, Appendix A.

### 4.8. Western Blotting

The Western blotting was performed as described previously [27]. The details of Western blotting are described in the Appendix A.

### 4.9. Histology and Immunohistochemistry

The hematoxylin and eosin (H&E) and immunohistochemistry (IHC) staining was performed according to routine protocols. The details of staining are described in the Appendix A.

### 4.10. Statistical Analysis

Data were statistically expressed using mean ± SD and one-way analysis of variance (ANOVA) followed by Tukey’s multiple comparison test was used for the statistical analysis of various three groups. *p* < 0.05 was considered statistically significant.

## 5. Conclusions

The present study links Twist and DDR proteins (PARP1 and XRCC1) with cisplatin resistance in OC. We demonstrated that Twist knockdown CisR OC cells are highly sensitive to cisplatin and PARPi niraparib in both in vitro 2D and 3D cell culture models (Figure 11). Our study also suggests that targeting Twist and PARPi (a synthetic lethal partner) could potentially be a beneficial approach to overcoming cisplatin-based resistance and a promising option for the treatment of OC.

## Figures and Tables

**Figure 1 ijms-22-03916-f001:**
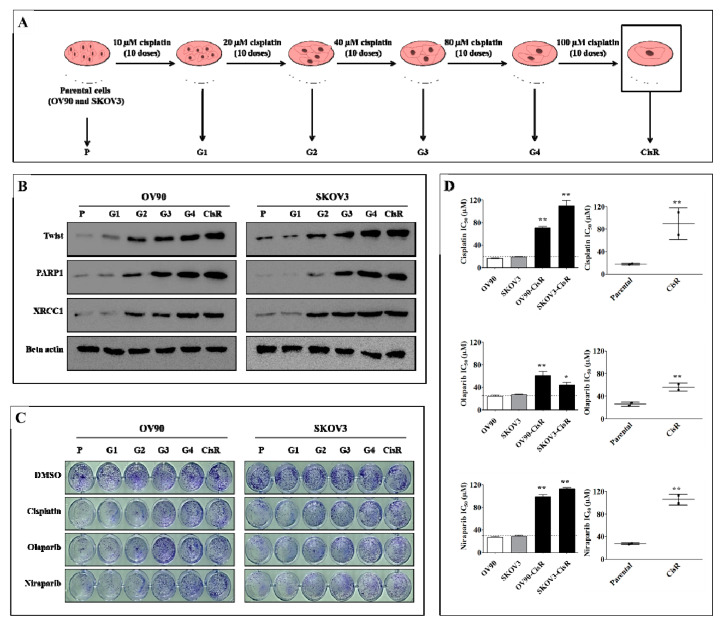
Twist expression regulates DNA damage response (DDR) proteins, PARP1 and XRCC1 during the development of cisplatin resistance in ovarian cancer (OC) cells. (**A**) Schematic repre-setantion of how four intermediate sublines were generated between parental and CisR OC cells. (**B**) Western blot analysis of Twist and DDR proteins, PARP1 and XRCC1 in each subline of OV90-CisR and SKOV3-CisR. (**C**) Clonogenic cell growth assay for cisplatin, olaparib and niraparib with the doses of 20 µM, each respectively, for 72 h of treatment followed by 72 h recovery in each subline of OV90-CisR and SKOV3-CisR. (**D**) Measurement of 50% inhibitory concentration (IC_50_) values of cisplatin, olaparib and niraparib on CisR OC cells compared with their respective parental cells. P: parental cell; CisR: cisplatin-resistant cells; G: sublines of each respective generation. Values were represented as mean ± SD. * *p* < 0.05, ** *p* < 0.01, compared with their respective parental cells.

**Figure 2 ijms-22-03916-f002:**
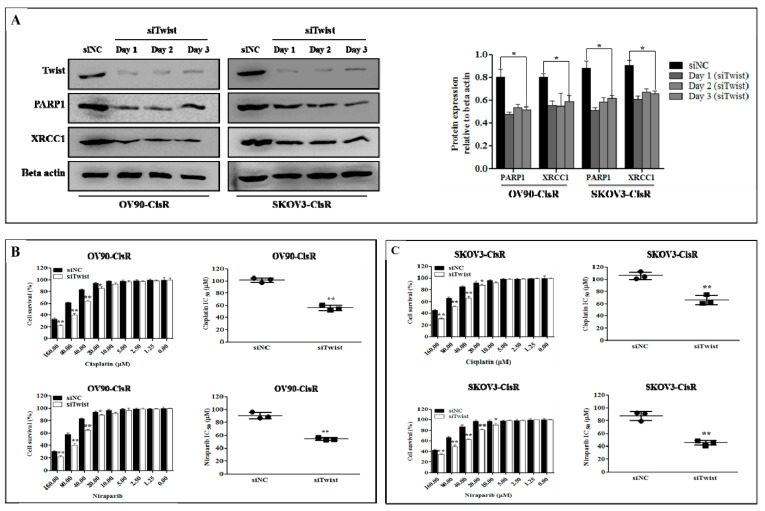
Twist regulates DDR proteins to sensitize cisplatin and PARPi in CisR OC cells. (**A**) Western blot analysis of Twist-knockdown with PARP1 and XRRRC1 proteins expression at day 1, 2 and 3 in OV90-CisR and SKOV3-CisR compared with the siNC group. (**B**,**C**) The treatment effect expressed as cell survival and IC_50_ of cisplatin and PARPi (niraparib) on CisR OV90 and SKOV3 OC cells depending on Twist expression. The both drus were treated at a various concentrations ranging from 0 to 160 µM for 72 h of incubation. CisR: cisplatin-resistant; siRNA: small interfering RNA; siNC: non-targeting negative control siRNA; siTwist: Twist siRNA. Values were represented as mean ± SD. * *p* < 0.05, ** *p* < 0.01 compared with the siNC group.

**Figure 3 ijms-22-03916-f003:**
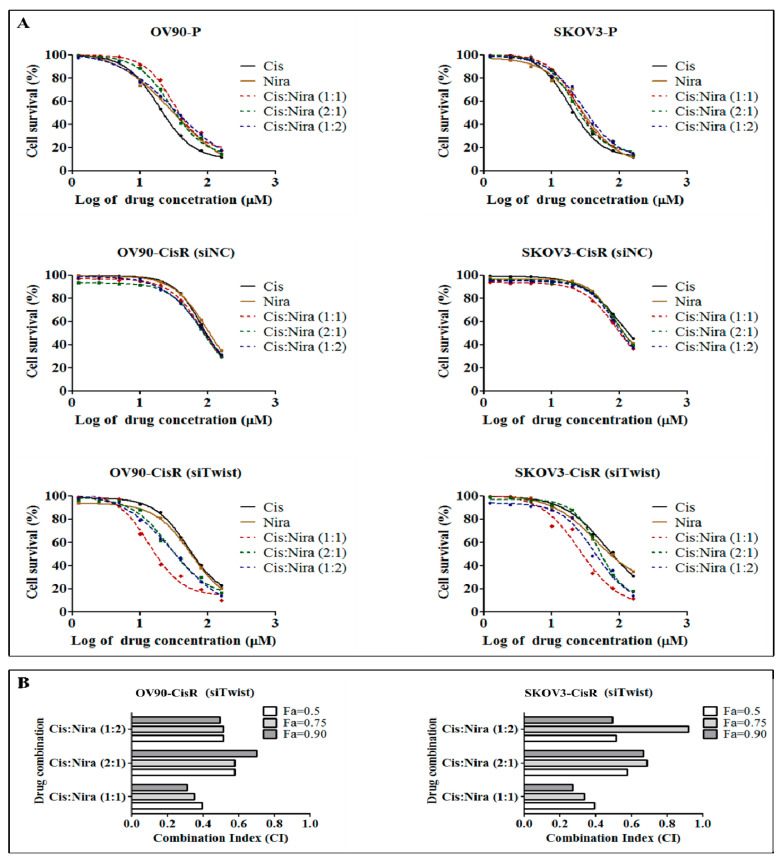
Twist knockdown sensitizes CisR OC cells to Cis and niraparib (Nira) therapy. (**A**) The dose response curve of Cis and Nira at 1:1, 2:1 and 1:2 combination ratio on parental, siNC CisR and siTwist CisR OV90 and SKOV3 cells. The cells were co-treated with increase dose of Cis and Nira (0–160 µM) for 72 h and percentage of cell survival was determined by EZ-cytox cell viability kit. (**B**) Measurement of combination index (CI) using compusyn software. P: parental CisR: cisplatin-resistant; siRNA: small interfering RNA; siNC: non-targeting negative control, siRNA; siTwist: Twist siRNA; DMSO: dimethyl sulfoxide, Cis: cisplatin; Nira; niraparib; Fa: fraction of affected cells.

**Figure 4 ijms-22-03916-f004:**
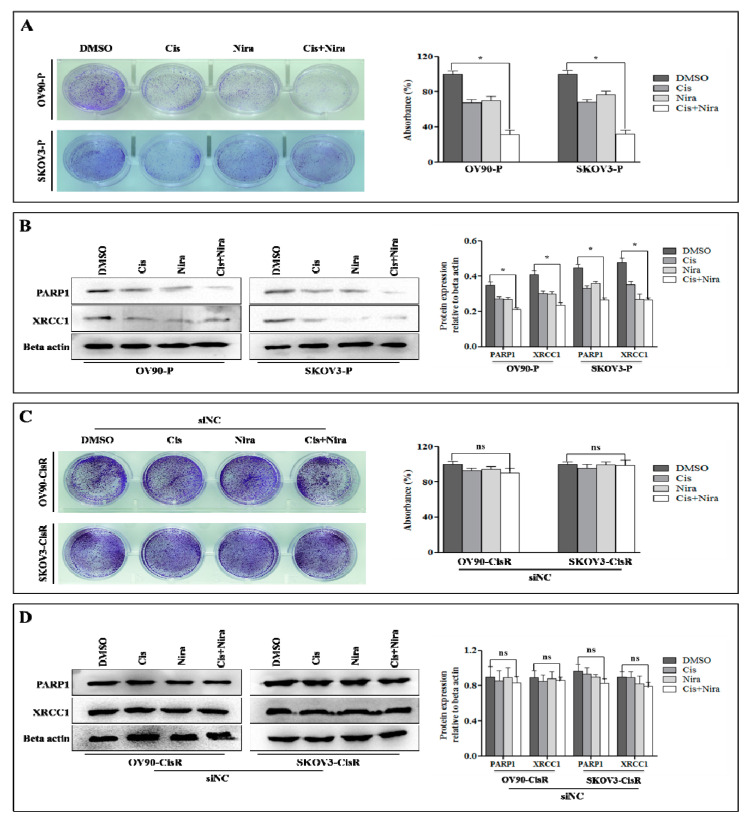
The effect of Cis and Nira on DDR proteins, PARP1 and XRCC1 in parental and non-silenced CisR OC cells. (**A**) Measurement of cell growth rate using clonogenic assay in parental cells administered with DMSO, Cis, Nira, and Cis+Nira. The cells were cultured in absence or presence of Cis (18 µm, OV90—P; 21 µm, SKOV3-P) and Nira (17.5 µm, OV90—P; 15 µm, SKOV3-P) for 72 h followed by 72 h of recovery period. (**B**) Western blot analysis of DDR proteins, PARP1 and XRCC1 expression levels to determine the effect on parental and siNC cells treated with DMSO, Cis, Nira and Cis+Nira. The transfected cells were cultured in absence or presence of Cis (18 µm, OV90—P; 21 µm, SKOV3-P) and Nira (17.5 µm, OV90—P; 15 µm, SKOV3-P) for 72 h. (**C**) Measurement of cell growth rate using clonogenic assay in siNC transfected CisR cells administered with DMSO, Cis, Nira, and Cis+Nira. The cells were cultured in absence or presence of Cis (18 µm, OV90—CisR; 21 µm, SKOV3-CisR) and Nira (17.5 µm, OV90—CisR; 15 µm, SKOV3-CisR) for 72 h followed by 72 h of recovery period. (**D**) Western blot analysis of DDR proteins, PARP1 and XRCC1 expression levels to determine the effect on parental and siNC cells treated with DMSO, Cis, Nira and Cis+Nira. The transfected cells were cultured in absence or presence of Cis (18 µm, OV90—CisR; 21 µm, SKOV3-CisR) and Nira (17.5 µm, OV90—CisR; 15 µm, SKOV3-CisR) for 72 h. P: parental; CisR: cisplatin-resistant; siRNA: small interfering RNA; siNC: non-targeting negative control siRNA; siTwist: Twist siRNA. Values were represented as mean ± SD. * *p* < 0.05, compared with the DMSO group and ^ns^
*p* > 0.05, compared with the DMSO group.

**Figure 5 ijms-22-03916-f005:**
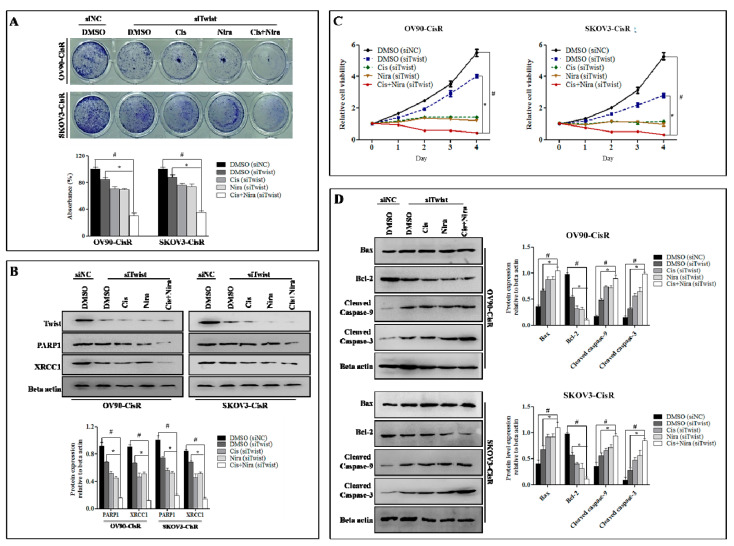
The Cis+Nira therapy induces cell death with blocking DDR capacity on Twist-knockdown CisR OC cells. (**A**) Measurement of cell growth rate using clonogenic assay in siTwist transfected cells administered with DMSO, Cis, Nira, and Cis+Nira. The siTwist transfected cells were cultured in absence or presence of Cis (18 µm, OV90—CisR; 21 µm, SKOV3-CisR) and Nira (17.5 µm, OV90—CisR; 15 µm, SKOV3-CisR) for 72 h followed by 72 h of recovery period. (**B**) Western blot analysis of DDR proteins, PARP1 and XRCC1 expression levels to determine the effect of siNC and siTwist transfected cells treated with DMSO, Cis, Nira and Cis+Nira. The transfected cells were cultured in absence or presence of Cis (18 µm, OV90—CisR; 21 µm, SKOV3-CisR) and Nira (17.5 µm, OV90—CisR; 15 µm, SKOV3-CisR) for 72 h. (**C**) Measurement of relative cell viability in siNC and siTwist transfected cells administered with DMSO, Cis, Nira, and Cis+Nira in OV90-CisR (18 µm of Cis and 17.5 µm of Nira) and SKOV3-CisR (21 µm of Cis and 15 µm of Nira) OC cells for 4 days. (**D**) Western blot analysis of cell death proteins, including Bax, cleaved caspase-9 and cleaved caspase-3 and Bcl-2 proteins and their expression levels to determine the effect of siNC and siTwist transfected cells treated with DMSO, Cis, Nira and Cis+Nira. The transfected cells were cultured in absence or presence of Cis (18 µm, OV90—CisR; 21 µm, SKOV3-CisR) and Nira (17.5 µm, OV90—CisR; 15 µm, SKOV3-CisR) for 72 h. CisR: cisplatin-resistant; siRNA: small interfering RNA; siNC: non-targeting negative control siRNA; siTwist: Twist siRNA. Values were represented as mean ± SD. # *p* < 0.05, compared with siNC group; * *p* < 0.05, compared with the siTwist (DMSO) group.

**Figure 6 ijms-22-03916-f006:**
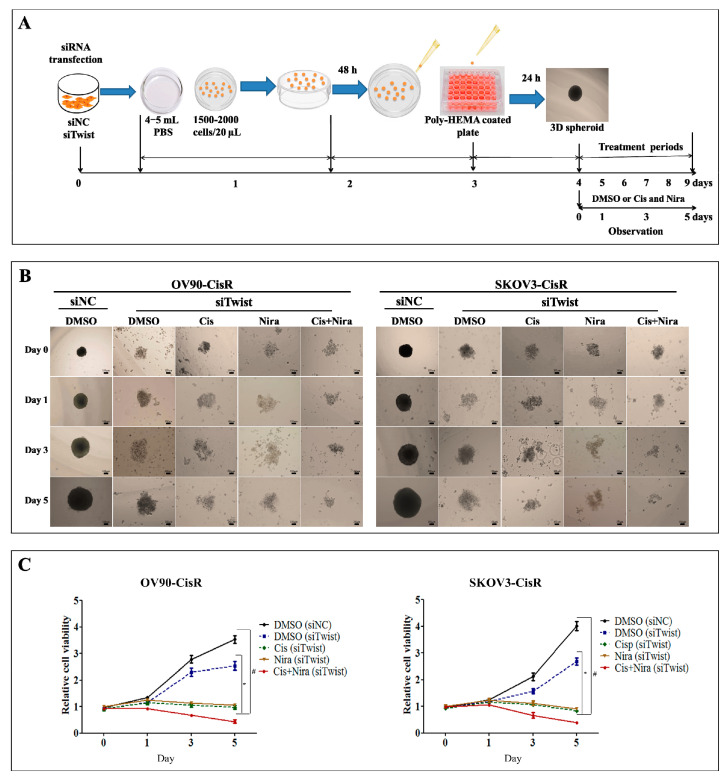
The Cis+Nira therapy is effective against three-dimensional (3D) cultures of Twist-knockdown CisR OC cells. (**A**) The schematic diagram showing the experimental protocol for the formation 3D spheroid and treatment plan of DMSO, Cis, and Nira. (**B**) The images representing 3D spheroids formation of siNC and siTwist transfected cells at 0, 1, 3 and 5 days on OV90-CisR and SKOV3-CisR (Magnification, 10×, scale bar 100 µm). The cells were cultured in absence or presence of Cis (18 µm, OV90—CisR; 21 µm, SKOV3-CisR) and Nira (17.5 µm, OV90—CisR; 15 µm, SKOV3-CisR) for 120 h. (Magnification, 10×, scale bar 100 µm) (**C**) Relative cell viability of siNC and siTwist transfected 3D culture cells administered with DMSO, Cis, Nira, and Cis+Nira in OV90-CisR (18 µm of Cis and 17.5 µm of Nira) and SKOV3-CisR (21 µm of Cis and 15 µm of Nira) OC cells. CisR: cisplatin-resistant; siRNA: small interfering RNA; siNC: non-targeting negative control, siRNA; siTwist: Twist siRNA; DMSO: dimethyl sulfoxide, Cis: cisplatin; Nira: niraparib. Values were represented as mean ± SD. # *p* < 0.05, compared with siNC group; * *p* < 0.05, compared with the siTwist (DMSO) group.

**Figure 7 ijms-22-03916-f007:**
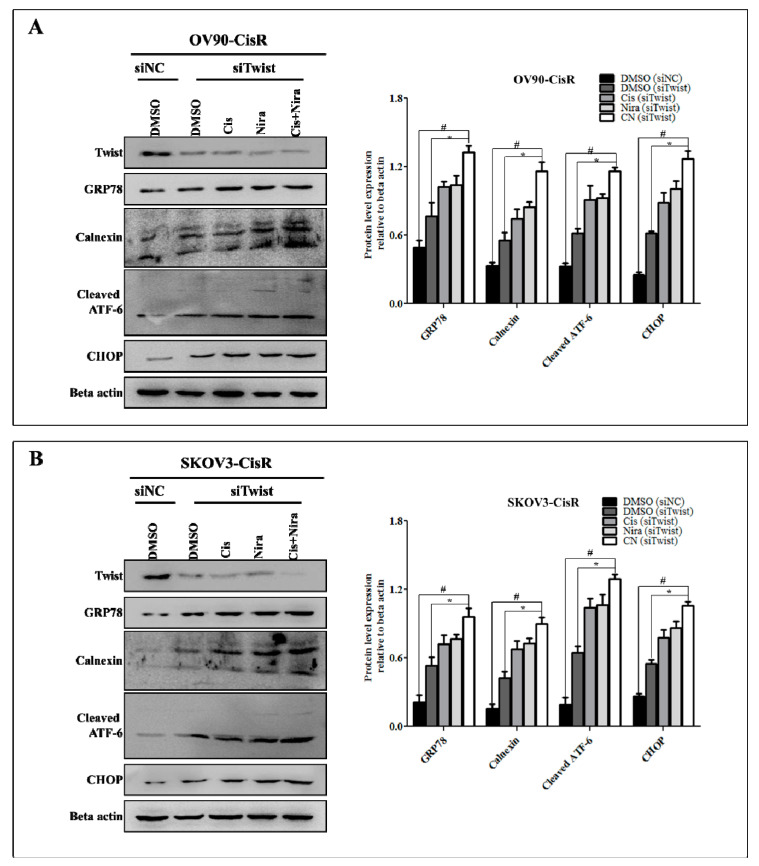
The Cis+Nira therapy enhances ER stress in 3D cultures of Twist-knockdown CisR OC cells. Western blot analysis of ER stress proteins, GRP78, calnexin, cleaved ATF6 and CHOP in 3D cultures of Twist-knockdown (**A**) OV90-CisR and (**B**) SKOV3-CisR OC cells. CisR: cisplatin-resistant; siRNA: small interfering RNA; siNC: non-targeting negative control, siRNA; siTwist: Twist siRNA; DMSO: dimethyl sulfoxide, Cis: cisplatin; Nira; niraparib. Values were represented as mean ± SD. # *p* < 0.05, compared with siNC group; * *p* < 0.05, compared with the siTwist (DMSO) group.

**Figure 8 ijms-22-03916-f008:**
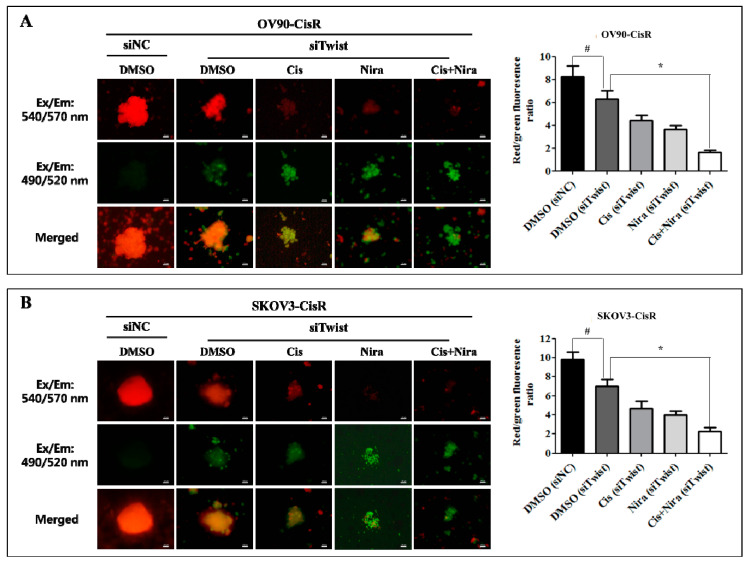
The Cis+Nira therapy significantly reduced mitochondrial membrane potential in Twist-knockdown CisR OC cells. Microscopic visualization and quantification of mitochondrial membrane potential in 3D cultures of Twist-knockdown (**A**) OV90-CisR and (**B**) SKOV3-CisR OC cells. Magnification, 10×, scale bar 100 µm. CisR: cisplatin-resistant; siRNA: small interfering RNA; siNC: non-targeting negative control, siRNA; siTwist: Twist siRNA; DMSO: dimethyl sulfoxide, Cis: cisplatin; Nira; niraparib. Values were represented as mean ± SD. # *p* < 0.05, compared with siNC group; * *p* < 0.05, compared with the siTwist (DMSO) group.

**Figure 9 ijms-22-03916-f009:**
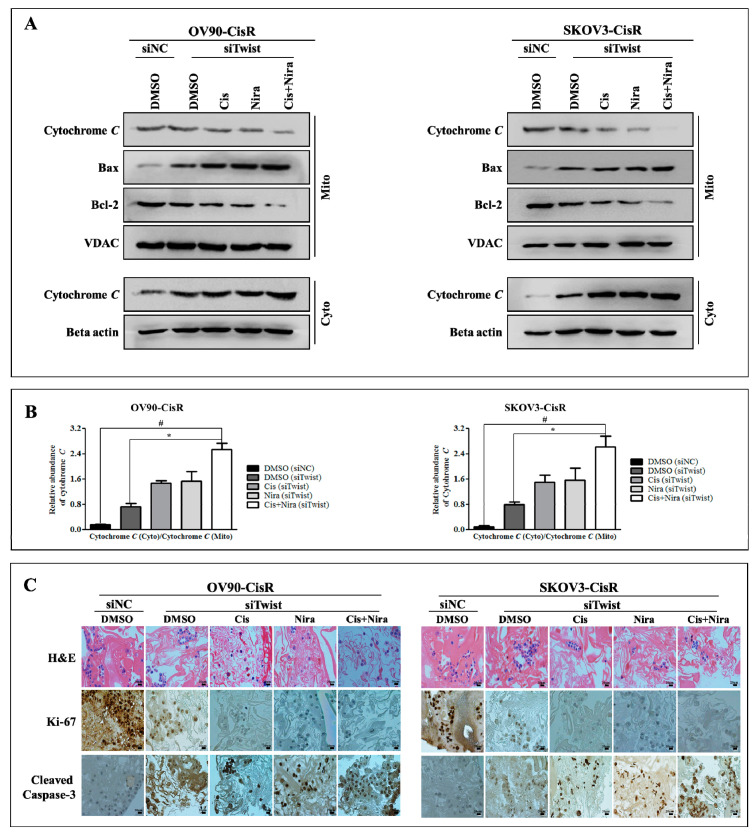
The combination therapy efficiently activated mitochondrial-dependent apoptotic pathway in Twist-knockdown CisR OC cells. (**A**) Western blot analysis of Bax, Bcl-2 and cytochrome C in 3D cultured of Twist knockdown OC cells. (**B**) The ratio of cytochrome C in cytosolic fraction to that in mitochondria fraction. (**C**) H&E staining for morphology evaluation and immunohistochemistry of the cell proliferation marker, ki-67 and apoptosis marker, cleaved caspase-3 immunoreactivity prediction (Magnification, 40×, scale bar 20 µm). CisR: cisplatin-resistant; siRNA: small interfering RNA; siNC: non-targeting negative control, siRNA; siTwist: Twist siRNA; DMSO: dimethyl sulfoxide, Cis: cisplatin; Nira; Niraparib. Values were represented as mean ± SD. # *p* < 0.05, compared with siNC group; * *p* < 0.05, compared with the siTwist (DMSO) group.

**Figure 10 ijms-22-03916-f010:**
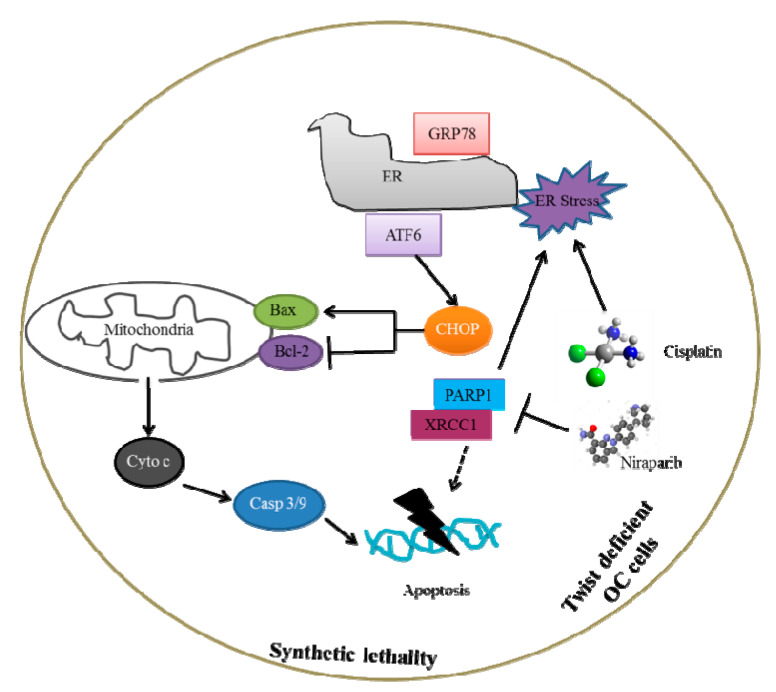
The pathway involved in the synthetic lethality of PARPi with combination of cisplatin on Twist knockdown CisR OC cells.

**Figure 11 ijms-22-03916-f011:**
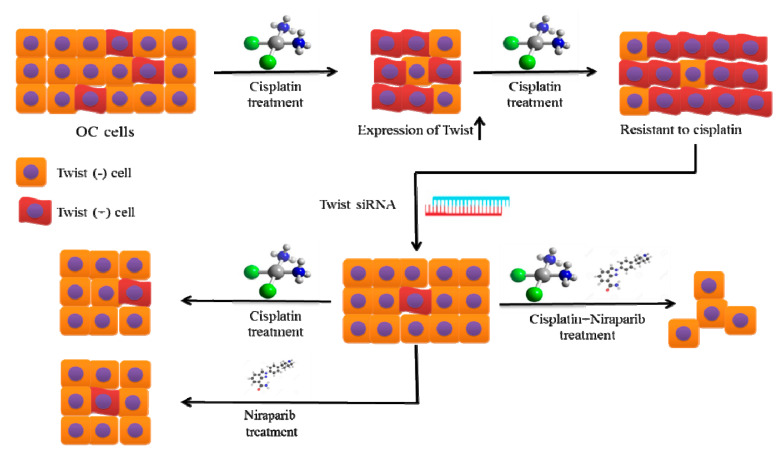
The proposed therapeutic strategy to overcome cisplatin resistance in OC.

## Data Availability

The data presented in this study are available in this article Int. J. Mol. Sci. and its Appendix A and methods.

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
