# Peer review of "Combination of Niraparib, Cisplatin and Twist Knockdown in Cisplatin-Resistant Ovarian Cancer Cells Potentially Enhances Synthetic Lethality through ER-Stress Mediated Mitochondrial Apoptosis Pathway"

_ijms, 2021, doi:10.3390/ijms22083916_

Round 1

Reviewer 1 Report

The main criticism is focused on the discrepancy between the obtained results and the conclusions reached by the authors.

  1. The doubling time of XO90 and SV30 cells is in range from 40 to 44 hours (according to the literature). So after this time period the effects Twist silencing should be gradually attenuated. How do the authors explain that even after 3 days, is the effect of Twist silencing so significant? (Fig. 2A). Does doubling time prolonged in cisR cell lines as compared with parental cells?
  2. Cis + Nira combination therapy suppresses cell growth (Fig. 3A) and reduces the level of PARP1 and XRRC1 proteins (Fig. 3C) in CisR Twist-knockdown cells. However, I lacked control experiments, which will bring data about effectiveness this therapy on non-silenced CisR cells and on original parental cells. Moreover, the expression levels of PARP1 and XRRC1 in all three cell variants under the influence of combination therapy should be measured.

3.   In the conclusions, the authors wrote: „We demonstrated that combination of Cis and Nira on Twist-knockdown OC cell could improve ovarian patient outcomes. Therefore, the combination strategy with Cis and Nira could be a secondary treatment option for patients who have developed Twist-dependent resistance to platinum-based chemotherapy“. This cannot be stated because authors bring evidence that this therapy is effective only on Twist silenced cells, and patients cannot be silenced prior treatment. Please do not use term: "ovarian patient outcomes", but "ovarian cancer patient’s outcomes".

  1. Captions for Figures should provide complete information about the experiments. Please include the data about the incubation period with the tested substances and the concentration of the substances used, not only their mutual ratios. It is difficult for the reader to estimate the concentration data, especially in the figures where you use a logarithmic scale.

Taken together, all the recommendations suggest that the authors have two options.  They can stick to the original conclusions and then they must fundamentally supplemented their paper experimentally. Otherwise they can change the conclusions that cisPt resistance in the cells used is associated with overexpression of Twist, PARP1 and XRRC1 and reorganize work accordingly.

Author Response

    1. The doubling time of XO90 and SV30 cells is in range from 40 to 44 hours (according to the literature). So after this time period the effects Twist silencing should be gradually attenuated. How do the authors explain that even after 3 days, is the effect of Twist silencing so significant? (Fig. 2A). Does doubling time prolonged in cisR cell lines as compared with parental cells?

    Reply: The confirmation of siTwist transfection efficacy was measured by western blot analysis on days 1, 2, and 3. We found that the Twist silencing remains steady until day 3 (PMID: 29658609). Our previous finding reported that The Twist deficient cells showed lower cell proliferation compare with non-silenced cells (PMID: 33076245). Also, some studies reported that the transfected cells exhibited a longer doubling time, lower plating efficiency, and a lower rate of DNA synthesis than the parental cells (PMID: 10814877).

    1. Cis + Nira combination therapy suppresses cell growth (Fig. 3A) and reduces the level of PARP1 and XRRC1 proteins (Fig. 3C) in CisR Twist-knockdown cells. However, I lacked control experiments, which will bring data about effectiveness this therapy on non-silenced CisR cells and on original parental cells. Moreover, the expression levels of PARP1 and XRRC1 in all three cell variants under the influence of combination therapy should be measured.

    Reply: Thank you very much for your valuable comments and suggestion that help us to improve manuscript quality. We have done additional experiments to include control groups according to reviewer suggestions. Please see Figures 2, 3, and 4. Lines 136-141, 151-155.

    1. In the conclusions, the authors wrote: „We demonstrated that combination of Cis and Nira on Twist-knockdown OC cell could improve ovarian patient outcomes. Therefore, the combination strategy with Cis and Nira could be a secondary treatment option for patients who have developed Twist-dependent resistance to platinum-based chemotherapy“. This cannot be stated because authors bring evidence that this therapy is effective only on Twist silenced cells, and patients cannot be silenced prior treatment. Please do not use term: "ovarian patient outcomes", but "ovarian cancer patient’s outcomes".

    Reply: We have modified the conclusion section. Please see lines 437-442.

    1. Captions for Figures should provide complete information about the experiments. Please include the data about the incubation period with the tested substances and the concentration of the substances used, not only their mutual ratios. It is difficult for the reader to estimate the concentration data, especially in the figures where you use a logarithmic scale.

    Reply: We have included the information regarding sample, concentration and incubation time in the figures.

    Taken together, all the recommendations suggest that the authors have two options. They can stick to the original conclusions and then they must fundamentally supplemented their paper experimentally. Otherwise they can change the conclusions that cisPt resistance in the cells used is associated with overexpression of Twist, PARP1 and XRRC1 and reorganize work accordingly.

    According to the reviewer's suggestion, we have revised our conclusion as “The present study links Twist and DDR proteins (PARP1 and XRCC1) with cisplatin resistance in OC. We demonstrated that Twist knockdown CisR OC cells are highly sensitive to cisplatin and PARPi niraparib in both in vitro 2D and 3D cell culture models. Our study also suggests that targeting Twist and PARPi (a synthetic lethal partner) could potentially be a beneficial approach to overcoming cisplatin-based resistance and a promising option for the treatment of OC.

Reviewer 2 Report

The authors demonstrated that combination of Cis and Nira on Twist-knockdown OC cell could improve ovarian patient outcomes and the combination strategy with Cis and Nira could be a secondary treatment option for patients who have developed Twist-dependent resistance to platinum-based chemotherapy. This report is very interesting.

Author Response

Thank you very much for your comments.

Round 2

Reviewer 1 Report

I note that the work has been modified according to my comments, I have no further comments.